# Comparison of Cultures and 16S/18S Amplicon-Based Microbiome Analyses for Diagnosing Nosocomial Pneumonia in Patients Admitted to the Intensive Care Unit—An Exploratory Study

**DOI:** 10.3390/diagnostics15243202

**Published:** 2025-12-15

**Authors:** Dennis Back Holmgaard, Lars Nebrich, Bülent Uslu, Christian Højte Schouw, Rimtas Dargis, Henrik Planck Pedersen, Kurt Fuursted, Henrik Vedel Nielsen, Jens Jørgen Christensen, Xiaohui Chen Nielsen, Lone Musaeus Poulsen

**Affiliations:** 1Department of Clinical Microbiology, Zealand University Hospital, 4200 Slagelse, Denmark; 2Department of Anesthesiology, Zealand University Hospital, 4600 Køge, Denmark; 3Department of Anesthesiology, Zealand University Hospital, 4000 Roskilde, Denmark; 4Department of Bacteria, Parasites, and Fungi, Statens Serum Institut, 2300 Copenhagen S, Denmark

**Keywords:** bacterial culturing, 16S/18S, microbiome, nosocomial pneumonia, intensive care unit

## Abstract

**Background**: Nosocomial pneumonia (NP) is a significant cause of morbidity and mortality in intensive care unit (ICU) patients. Prior antibiotic use, polymicrobial infections, and the limitations of conventional microbiological methods often complicate an accurate diagnosis. Bronchoalveolar lavage (BAL) and tracheal suction (TS) are commonly used methods for collecting respiratory samples; however, their diagnostic accuracy can vary. Additionally, microbiome analysis using 16S/18S rRNA gene sequencing provides an alternative approach for identifying pathogens that are difficult to culture. This study aimed to compare the diagnostic value of routine culturing and microbiome analysis in identifying pathogens in ICU patients with NP. **Methods**: A prospective cohort study was conducted in 23 critically ill patients at Zealand University Hospital. Samples from TS and BAL were collected from patients with suspected NP. Both culturing and 16S/18S rRNA gene amplicon-based microbiome analysis were performed to identify pathogens. Findings were compared between the two types of samples and between the two analysis methods. **Results**: A total of 46 samples were analyzed (23 TS and 23 BAL). Culture results showed complete concordance in 60.9% of cases and partial concordance in 21.7% between results from TS and BAL. Discrepancies often involved low-virulence organisms, such as *Staphylococcus epidermidis* and *Candida albicans*. Microbiome analysis revealed a broader spectrum of microbial diversity, detecting pathogens such as *Pasteurella canis* and *Tropheryma whipplei* that were previously missed by culture methods. In 34.8% of the samples, the pathogen identified by microbiome analysis was also detected by culture. However, microbiome analysis also identified additional microorganisms in 17.4% of the cases, which were not detected by culture. When comparing microbiome results between TS and BAL, 16 out of 23 (69.5%) showed complete concordance. **Conclusions**: The findings were similar in TS and BAL, both for culture and 16S/18S amplicon-based microbiome analyses. Microbiome analysis using 16S/18S rRNA gene sequencing provided new insights into NP patients, identifying pathogens that were previously undetected by conventional culturing methods. Combining microbiome analysis with traditional culture techniques could enhance the diagnostic accuracy for NP. Further studies are needed to refine diagnostic thresholds and assess the clinical impact of microbiome-based diagnostics.

## 1. Introduction

Pneumonia is the most common type of nosocomial infection in patients admitted to intensive care units (ICUs) [1]. A prospective multi-center cohort study conducted in nine European countries in 2009 found that up to 76% of all cases of pulmonary infections could be attributed to nosocomial causes [2]. Nosocomial pneumonia (NP) is associated with increased ICU length of stay, longer duration of mechanical ventilation, and higher mortality [2,3]. NP is caused by a variety of bacterial pathogens, with different resistance patterns [4]. Establishing a microbiological diagnosis is crucial for targeted narrow-spectrum antibiotic therapy, but it is complicated by several factors, such as the availability of respiratory specimens and the viability of the microorganisms present in the specimen [4].

Routine tracheal suction (TS) from the endotracheal tube, or more invasive procedures such as bronchoalveolar lavage (BAL), are typically used to collect samples. The latter is usually reserved for patients experiencing treatment failure or when exclusion of other differential diagnoses is necessary [5]. This conservative approach stems from the risk of complications associated with BAL, such as respiratory and circulatory instability or bleeding [6]. While sampling from the lower airways theoretically offers a more representative cause of infection, a Cochrane meta-analysis failed to demonstrate added benefits of BAL for routine diagnostics in patients with ventilator-associated pneumonia (VAP) [7].

The role of BAL in NP remains a topic of debate [8,9]. The latest guidelines from the Infectious Disease Society of America (IDSA) recommend tracheal sampling for the diagnosis of hospital-associated pneumonia (HAP) and VAP [8]. In contrast, the European guidelines advocate for distal sampling before antibiotic treatment to improve diagnostic accuracy and minimize the use of unnecessary antibiotics [9]. Both recommendations are supported by “low-quality” evidence, highlighting the need for more research in this area.

Prior antibiotic use before ICU admission further complicates microbiological diagnosis and is a common occurrence as antibiotics are usually started before admission to the ICU, leading either to the usage of empiric treatment with broad-spectrum antibiotics or the premature cessation of antibiotics due to culture negative samples. Conventional methods can establish a microbial diagnosis in 70% of all cases in intubated patients with pneumonia, and in 30% of all cases, more than one bacterium is present [10]. Culturing bacteria depends on the viability of microorganisms, which can be affected by prior antibiotic treatment. Microbiome analyses, based on sequencing nuclear ribosomal genes (16S and 18S) amplified from genomic DNA using Next Generation Sequencing (NGS), could help bypass this limitation, as the presence of viable bacteria is no longer required for detection. Furthermore it could increase the detection of fastidious bacteria that would not be found using conventional culturing methods. Therefore, next-generation sequencing has been increasingly used to detect pathogens in clinical samples, particularly for the diagnosis of lower respiratory tract infections [11,12].

This study aimed to characterize the microbiome of samples of routine TS and compare it to samples collected via BAL obtained from intubated patients with nosocomial pneumonia admitted to the ICU. These samples were investigated with 16S/18S amplicon-based microbiome analyses to assess whether this provided more accurate microbiological diagnoses and to evaluate whether combining culture and microbiome analysis from TS and BAL provided a more precise diagnostic approach for identifying the etiological agents of NP in intubated patients with nosocomial pneumonia admitted to the ICU.

## 2. Materials and Methods

### 2.1. Study Design

This study was a prospective, descriptive cohort study conducted in intubated, critically ill patients with nosocomial pneumonia admitted to the ICUs at Zealand University Hospital in Roskilde and Køge.

### 2.2. Population

Patients were included if they met the following criteria: age ≥ 18 years, admission to the ICU, requiring endotracheal intubation, and a diagnosis of NP. The diagnosis of NP was established according to the Centers for Disease Control and Prevention (CDC) criteria.

The CDC criteria for pneumonia diagnosis consist of the presence of a new or progressive radiographic parenchymal lung infiltrate, accompanied by at least two of the following: temperature alterations (below 36 °C or above 38.0 °C), white blood cell count < 4000 cells/mm^3^ or >12,000 cells/mm^3^, or purulent-appearing airway samples [13]. For the pneumonia to be categorized as nosocomial, the onset should be at least 48 h after hospital admission [14].

The exclusion criteria were: terminal illness, lung cancer, being intubated for more than 24 h before screening, or not being compliant with the CDC criteria for pneumonia. The limit to the length of intubation was based on the risk of biofilm formation in the endotracheal tube and risk of including colonization in the analyses [15]

### 2.3. Data Collection

At baseline, the following demographic data were recorded: age, sex, admission diagnosis, ICU length of stay (LOS), duration of mechanical ventilation, Simplified Acute Physiology Score (SAPS3), C-reactive protein (CRP), Leukocyte count, and type of antibiotic treatment before and after sample collection.

### 2.4. Sample Collection

For each patient, samples were obtained at the time of inclusion in the study via routine TS followed by mini-bronchoalveolar lavage (mini-BAL). The mini-BAL was performed using a fiberoptic bronchoscope (Ambu^®^ aScope™ 3 Broncho Regular 5.0/2.2. Ambu A/S, 2750 Ballerup, Denmark). Normal saline was instilled into the lung with a radiologically verified new infiltration, or, in cases of uncertainty, into the right lung. The applied volume of normal saline ranged from 20 to 40 mL. Both procedures were performed through an endotracheal tube already in place due to the need for mechanical ventilation. All samples were divided into two. One sample was sent for conventional culturing and another underwent microbiome analysis. The samples were sent and stored in accordance with local guidelines for microbiological samples. The samples were kept at 4 °C until analyzed and afterwards stored in long-term storage at −80 °C.

### 2.5. Culturing Samples

All samples were seeded onto 5% blood agar plates (SSI Diagnostica, Hillerød, Denmark), chocolate agar plates (SSI Diagnostica, Hillerød, Denmark), 1/3 chromogenic blue agar plates (SSI Diagnostica, Hillerød, Denmark), 1/3 tellurite-containing agar plates (SSI Diagnostica, Hillerød, Denmark), anaerobic agar plates (SSI Diagnostica, Hillerød, Denmark), and chromogenic Colorex Candida agar plates (SSI Diagnostica, Hillerød, Denmark). The plates were incubated at 37 °C for 20 to 40 h. After incubation, all colonies were isolated and identified using MALDI-TOF MS (Bruker, Germany). All bacterial findings, regardless of their clinical relevance, were reported to facilitate an accurate comparison of microbiological results.

### 2.6. Microbiome Analysis (16S/18S rRNA Gene Amplicon Sequencing)

#### 2.6.1. DNA Extraction, PCR and Sequencing

Both TS and BALs were analyzed using a combined 16S/18S DNA gene amplicon-based microbiome analysis to identify species-specific sequences. Briefly, 1 mL sample was disrupted on a TissueLyser II instrument (Qiagen, Hilden, Germany) with 1.4 mm Zirconium Beads (OPS Diagnostics, Lebanon, NJ, USA) at 30 Hz for 2 min, followed by 3000× *g* centrifugation at 3 min. The supernatant was used for Genomic DNA extraction by the NucliSENS^®^ easyMAG^®^ apparatus (bioMérieux, Marcy-l’Étoile, France) according to the manufacturer’s instructions. For each DNA extraction batch, a negative control without sample material was included for downstream analysis. DNA amplification was performed using a two-step PCR with modified versions of the published universal prokaryotic primers 341F (ACTCCTAYGGGRBGCASCAG) and 806R (AGC GTG GAC TAC NNG GGT ATC TAA T), targeting the V3–V4 regions of the 16S rRNA gene and three primer sets to target the eukaryotes namely, G3F1/G3R1 (5′-GCC AGC AGC CGC GGT AAT TC-3′/5′-ACA TTC TTG GCA AAT GCT TTC GCA G-3′), G4F3/G4R3 (5′-CAG CCG CGG TAA TTC CAG CTC-3′/5′-GGT GGT GCC CTT CCG TCA AT-3′), and G6F1/G6R1 (5′-TGG AGG GCA AGT CTG GTG CC-3′/5′-ACG GTA TCT GAT CGT CTT CGA TCC C-3′) [16]. Amplicons were sequenced on the Illumina MiSeq desktop sequencer (Illumina Inc., San Diego, CA, USA), using the V2 Reagent Kit.

#### 2.6.2. Data Analysis

Data analysis was performed using the BION pipeline (http://box.com/bion) (accessed on 22 April 2021). This pipeline handles raw sequences and includes several steps: de-multiplexing, primer extraction, sampling, sequence- and quality-based trimming and filtering, de-replication, clustering, chimera checking, reference data similarity, and taxonomic mapping and formatting. Non-overlapping paired reads were allowed for analysis. Before data analyzing, the BION output of four csv files, representing the four primer sets, were combined in one csv file.

In the absence of a reference methodology for NGS-based diagnostics of NP, a proportion-based approach was adopted to identify clinically relevant pathogens. Specifically, pathogens contributing more than 40% of the total sequence reads were considered clinically significant and reported.

This conservative approach ensures that only bacterial reads present at substantial levels were considered relevant, thereby increasing the likelihood of their involvement in NP. Species pathogens contributing fewer than 500 reads were excluded from the analysis.

### 2.7. Statistics

This study is regarded as exploratory and descriptive. No sample size calculation was performed. Descriptive statistics were applied. Data are reported as counts and proportions, and linear data as medians and interquartile ranges. All analyses and figures were carried out in R (R version 4.3.1, http://www.r-project.org) (accessed on 8 March 2023).

## 3. Results

A total of 23 patients were included in the study, and 46 samples were collected (23 BAL and 23 TS samples). Demographic and clinical characteristics of the patients are summarized in Table 1. The median age was 72 (IQR 65–75), 78% were male, and the patients had severe acute critical illness with SAPS2 of 74 (IQR 69–83). The median times on the ventilator and in the ICU were high: 11 days (IQR 5–20) and 13 days (IQR 12–25), respectively (Table 1). The admission type was surgical for 5/23 patients (21.7%), and 8/23 (34.8%) had a positive SARS-CoV-2 test result. All patients but one received antibiotics before ICU admission (Appendix A).

### 3.1. Culture

Complete concordance was observed between the BAL and TS samples in 14 out of 23 patients (60.9%) following routine cultivation. Partial concordance (without clinically significant discrepancies) was observed in 5 out of 23 patients (21.7%) (patients 3, 12, 13, 15, and 16) (Appendix A). In these cases, additional microorganisms were detected in the TS samples but not in the BAL, including *Staphylococcus epidermidis*, *S. haemolyticus*, and *Candida krusei*.

In two of the remaining patients, no growth was detected in the BAL samples (patients 2 and 10), while low-virulence potential pathogens were identified in the corresponding TS samples (Appendix A).

Conversely, in one patient (patient 20), bacteria of low virulence were detected in the BAL but not in the TS sample. Finally, in one patient (patient 14), different pathogens were cultured from both BAL and TS, although both were found at low levels, and none were considered likely to be the cause of pneumonia. These included *S. epidermidis*, *Neisseria* spp., and *Candida albicans*. Further details are provided in Appendix A.

### 3.2. Microbiome Analyses

#### 3.2.1. 16S-18S Sequence Read Output

16S and 18S rDNA data were generated for all samples, yielding between 35,419 and 577,008 reads per sample (median, 87,902; interquartile range [IQR], 63,230–115,957) (Figure 1). Vertebrate (host) DNA represented a substantial proportion of all sequences generated (median 71,190; IQR 57,201–85,514).

No DNA was recovered from the BAL sample of patient 2, and no DNA was recovered from the TS sample of patient 5 (Figure 1 and Appendix A). Culturing of these samples revealed low-level growth of potential contaminants (Appendix A).

In 22 samples, no microorganisms were identified. In 12 of these, no microbial DNA was detected, while in the remaining 10 samples, the microorganisms present did not meet the 40% abundance threshold for reporting (Figure 1 and Appendix A).

#### 3.2.2. Bacterial and Fungal Composition

The bacterial and fungal compositions from the 16S/18S amplicon-based microbiome analyses were compared between the BAL and TS samples (Figure 2 and Appendix A). Complete concordance was observed in 16 out of 23 cases (69.5%). Negative results were found in both samples in 8 cases. In the remaining 8 cases, a range of microorganisms was identified, including *Escherichia coli* (3 cases), *Klebsiella pneumoniae* (1 case), *Pasteurella canis* (1 case), *Staphylococcus epidermidis* (1 case), *Candida glabrata* (1 case), and *Tropheryma whipplei* (1 case) (Appendix A).

In three patients, microbiome analysis of the BAL was negative, but it identified microorganisms in the TS samples. In one patient, *Streptococcus pneumoniae* was detected in the BAL, but no microorganism was identified in the TS sample. In another patient (patient 1), two different species of *Candida* were identified in the BAL. In yet another case (patient 4), *Candida glabrata* was found in the BAL, while *S. haemolyticus* was identified in the TS. In patient 18, the BAL sample revealed the presence of *Pseudomonas fluorescens* and *Serratia marcescens*, while the TS sample contained *Haemophilus influenzae*.

### 3.3. Comparison Between Results Using Two Diagnostic Methods

Greater differences were observed when comparing the results obtained from the culturing and microbiome analysis (Figure 2 and Appendix A). In 16 out of 46 samples (34.8%), the dominant microorganism identified by microbiome analysis was also identified by culturing. In 8 out of 46 samples (17.4%), microbiome analysis detected microorganisms that were not identified by culturing. In the remaining 22 samples (47.8%), no microorganisms were identified by microbiome analysis.

In two cases, unusual pathogens were detected by microbiome analysis but not by traditional culturing methods. In one case (patient 9), *Pasteurella canis* was identified, accounting for more than 50% of the non-vertebrate DNA reads. In another case (patient 20), *Tropheryma whipplei* was identified, accounting for more than 95% of the non-vertebrate reads.

## 4. Discussion

In this study, we aimed to compare the microbiological diagnostic value of two different sampling methods—TS and BAL—and two diagnostic methods—routine culturing and 16S/18S rRNA gene amplicon-based microbiome analysis—in diagnosing the etiological pathogens in intubated, critically ill patients with NP in an ICU setting. Our findings suggest that while routine culturing of TS and BAL samples can provide equally valuable information, 16S/18S amplicon-based microbiome analysis may offer additional insights into microbial composition, which could aid in more precise diagnostic outcomes.

In our study, Gram-negative bacilli, such as *E. coli and K. pneumoniae,* were detected in 7 out of 23 patients by culture (Appendix A). This number is relatively low compared to other studies. Martin-Loeches et al. demonstrated in a recent multinational, prospective cohort study of nosocomial lower respiratory tract infections (LRTIs) that the most frequently identified microorganisms were Gram-negative bacteria, including *P. aeruginosa* (18.4% [186/1008]) and *Klebsiella* spp. (14.4% [145/1008]), *A. baumanii* (11.0% [111/1008]), and *E. coli* (8.5% [86/1008]). Gram-positive isolates were predominantly methicillin-susceptible *S. aureus* (MSSA) (10.81% [109/1008]) [17]. The lower detection rate of Gram-negative bacilli in our cohort may be attributed to the fact that all patients but one in this study had received antibiotic treatment before the onset of NP (Appendix A). Eight patients tested positive for SARS-CoV2. We observed no difference in the observed microbiomes between the group testing positive and the group testing negative although our study was not of a sufficient size to rule out such an association.

The concordance between TS and BAL samples in cultures was relatively high, with 82.6% of the samples showing concordant results (60.9% showing complete concordance and 21.7% showing partial concordance). This finding is consistent with other studies, which report a comparable diagnostic yield between these two sampling methods [18]. However, discrepancies were observed in several patients, particularly where low-virulence pathogens such as *Staphylococcus epidermidis* and *Candida albicans* were cultured. These organisms, often regarded as contaminants or opportunistic pathogens, highlight the challenges in diagnosing NP in critically ill patients, as the clinical relevance of such findings can be challenging to interpret, especially when they are present in low quantities. This is in line with previous studies that show difficulty in isolating clinically significant pathogens from samples in ICU patients due to prior antibiotic use or the presence of non-pathogenic organisms [19,20].

Interestingly, microbiome analysis showed concordant results compared with culture in most cases. However, it provided more nuanced insights into the microbial communities present, detecting a broader range of bacterial and fungal species, some of which were not identified by traditional culture methods. This was expected, as microbiome analysis does not rely on culturing viable microbiota. This finding aligns with earlier studies. For instance, Heitz et al. [21] analyzed 32 BAL fluids using culture and metagenomic next-generation sequencing (mNGS). Of these, 22 (69%) were positive by culture, and mNGS also reported all pathogens identified. Among the culture-positive BAL samples, additional bacterial species were revealed by mNGS for 12 patients. Among the BAL fluid with culture-negative test, five culture-negative BAL samples were positive by mNGS. As a shotgun-based, untargeted approach, mNGS can potentially provide a more comprehensive and unbiased overview of microbial composition compared with the 16S/18S amplicon-based strategy used in our study. However, our use of this amplicon-based strategy using primers designed to target pro- and non-mammalian eukaryotes may offer a more cost effective diagnostic sensitivity, using less sequencing capacity if relevant non-mammalian eukaryotes are present.

In our study, an unusual bacterium was found using microbiome analysis in two cases. In patient 9, both samples were positive for *Pasteurella canis* by NGS. Conventional culturing of both BAL and tracheal samples identified mixed flora, including *Staphylococcus aureus*, *Enterobacter cloacae*, *Escherichia coli*, *Candida tropicalis*, and *Candida albicans*. Despite the high levels of *P. canis* detected in the NGS analysis of both BAL and tracheal samples, the identification of multiple microorganisms may have interfered with the conventional culturing process, making it challenging to isolate *P. canis* using standard methods. Since the patient received piperacillin-tazobactam treatment to which *P. canis* is susceptible, this finding had no consequence for the antimicrobial therapy. In patient 20, *Tropheryma whipplei* was identified in both sample types by microbiome analyses, accounting for more than 90% of non-vertebrate reads. Conventional culturing showed weak growth of *S. epidermidis* in the BAL sample and no growth in the tracheal sample. This is not surprising, as routine culturing of airway material is not expected to find *Tropheryma whipplei*. The patient was undergoing treatment for suspected Crohn’s disease, and the first-line treatment had failed before ICU admission. Although the patient had undergone diagnostic procedures, no signs of Whipple’s disease were present. We assume that the presence of *Tropheryma whipplei* was barely colonization. Earlier studies of the lung microbiome using culture-independent techniques have shown that *T. whipplei* can be present in the lungs of healthy individuals, although the origin of this organism remains unclear [22]. In a study by Qin SL et al., *T. whipplei* was detected in BAL and gastric aspirate samples from healthy individuals using a *T. whipplei*-specific PCR assay [23].

Moreover, microbiome analysis was able to identify specific microbial sequences even in cases where conventional culturing yielded negative or ambiguous results. In patient 14, *S. pseudopneumoniae* was identified in the BAL by microbiome analysis, while in culture, there was weak growth of *Neisseria* spp. and Candida albicans in the BAL and TS cultures, respectively. *S. pseudopneumoniae* could potentially be the etiological agent for the pneumonia in this patient. The patient received piperacillin and tazobactam before and after the samples were taken; therefore, *S. pseudopneumoniae* could not be inhibited in culture. This finding highlights the potential of microbiome analysis to overcome some limitations of traditional culture methods, particularly in the context of ICU patients who may have polymicrobial infections or prior antibiotic exposure, which can limit bacterial viability [24,25].

While microbiome analysis offers promising advantages, such as the ability to detect microorganisms that are difficult to culture or that may be present in low abundance, its implementation is not without challenges. Whether the detected microorganisms are true causative agents of the pneumonia or merely part of the resident microbiome remains debatable [26]. Additionally, the absence of a definitive diagnostic threshold for microbiome-based diagnostics in NP, along with the need for further validation of microbiome findings, limits the clinical applicability of these results. Moreover, we applied a rather restrictive approach in defining potential pathogens, considering only those microbial sequences that accounted for more than 40% of non-vertebrate reads. This may limit our ability to investigate polymicrobial NP, and further studies are needed to determine whether this threshold is too restrictive. In most cases, one of the cultured pathogens was detected using microbiome analysis; however, multiple microorganisms were not identified.

The descriptive nature of the study, along with the low number of cases included, represents a limitation to the strength of the conclusions that can be drawn.

## 5. Conclusions

The findings are similar when comparing TS and BAL, both for culture and 16S/18S amplicon-based microbiome analyses. Microbiome analysis using 16S/18S rRNA gene sequencing provided additional insights into the microbial communities in NP patients, identifying pathogens that were undetected by conventional culturing methods. Combining microbiome analysis with traditional culture techniques could enhance the diagnostic accuracy for NP, offering potential for improved management and targeted antibiotic therapy in ICU patients. Further studies are needed to refine diagnostic thresholds and assess the clinical impact of microbiome-based diagnostics.

## Figures and Tables

**Figure 1 diagnostics-15-03202-f001:**
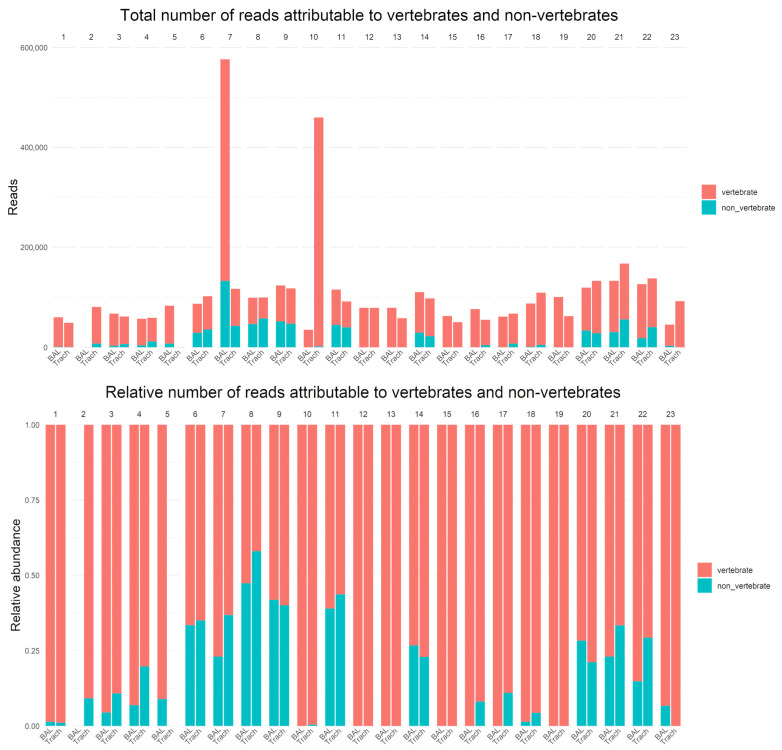
Total number of reads attributable to vertebrates and non-vertebrates (**top**). Relative number of reads attributable to vertebrate and non-vertebrate (**bottom**).

**Figure 2 diagnostics-15-03202-f002:**
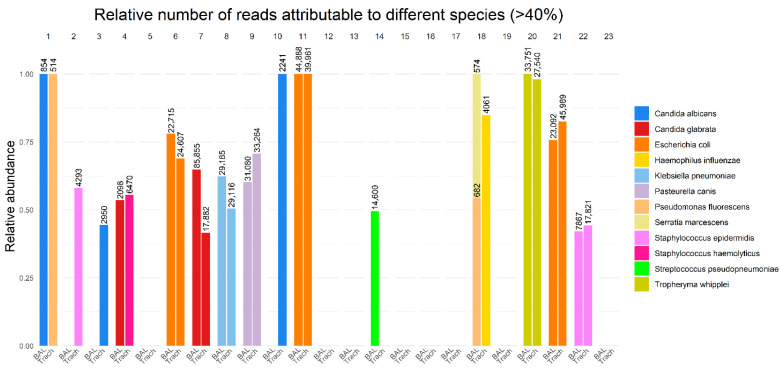
Total number of reads displayed as number above each sample and relative number of reads of the total number of reads attributable to a single pathogen in BAL and TS samples on the y-axis.

**Table 1 diagnostics-15-03202-t001:** Demographic data. Demographic data are either displayed as a median with and an (IQR) inter-quartile range or as a number displaying relative proportion in (%) percentages.

	N = 23
Age, median (IQR) years	72 (65–75)
Males, n (%)	18 (78)
ICU LOS, median (IQR) days	13 (12–25)
Admission type surgical, n (%)	5 (21,7)
Mechanical ventilation, median (IQR) days	11 (5–20)
CRP, median (IQR) mg/L	219 (157–287)
Leucocyte, median (IQR) 10^9^/L	12 (8–15)
SAPS3, median (IQR)	74 (69–83)
SARS-CoV 2 positive, n (%)	8 (34)

ICU: intensive care unit; LOS: Length of stay; CRP: C-reactive protein; SAPS3: Simplified Acute Physiology Score 3.

## Data Availability

The data presented in this study are available on request from the corresponding author.

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
