# Peer review of "Comparison of Cultures and 16S/18S Amplicon-Based Microbiome Analyses for Diagnosing Nosocomial Pneumonia in Patients Admitted to the Intensive Care Unit—An Exploratory Study"

_diagnostics, 2025, doi:10.3390/diagnostics15243202_

Round 1
Reviewer 1 Report
Comments and Suggestions for Authors
Dear Authors
Congratulations on a well done study,
some notes and questions;
Table 1
It would be of high interest if the antibiotic treatment that is stated in line 107(before and after the sampling) is also included. In addition it would be of high interest a discussion upon the treatment,microbiome analysis in accordance to lenght of stay.
sample collection
(110-116)- isn't it basic to report the exact time that the collection happened( eg the samples were collected during the 2nd day of NP diagnosis)?
sample size
(154) No sample size calculation was performed because of the exploratory nature of the study, but has data saturation been achieved?
Author Response
It would be of high interest if the antibiotic treatment that is stated in line 107 (before and after the sampling) is also included. In addition it would be of high interest a discussion upon the treatment,microbiome analysis in accordance to lenght of stay.
The data about antibiotic treatment is presented in supplementary table S1.
This is a prospective study and the cultures were performed “real time” after sampling. However, the microbiome based on NGS was performed more than 3 months after the samples were collected. Therefore, the results of the NGS analysis did not have influence on treatment.
(110-116)- isn't it basic to report the exact time that the collection happened(eg the samples were collected during the 2nd day of NP diagnosis)?
We added the information “the same day when NP is diagnosed” in the sample collection section (line 132)
(154) No sample size calculation was performed because of the exploratory nature of the study, but has data saturation been achieved?
This study is an exploratory and descriptive study. The aim was to describe findings from two diagnostic methods, not to assess statistical significance or generate thematic saturation. Consequently, the concept of data saturation does not apply, and a formal sample size calculation was neither feasible nor required for this study design.
Reviewer 2 Report
Comments and Suggestions for Authors
General review:
This is a very interesting and well-designed study on a topic of real clinical importance. The challenge of accurately diagnosing nosocomial pneumonia in the ICU is one we all grapple with, and your work comparing both sampling methods (TS and BAL) and diagnostic techniques (culture and microbiome analysis) in the same group of patients is a valuable contribution. I particularly appreciated the detailed discussion of the two unusual pathogens your team identified.
Additional suggestions:
- To immediately frame the scale of this work for the reader, you might consider adding the total number of patients (N=23) to the methods section of the abstract.
- The manuscript lacks any information regarding the handling of samples after collection. Key details such as the maximum time allowed between collection and processing, and the storage temperature (e.g., 4°C, -80°C) during that interval should be added.
- The protocol specifies a kit-based DNA extraction method but fails to mention whether a mechanical lysis step was included. Standard enzymatic lysis is notoriously inefficient for breaking open organisms with tough cell walls, particularly Gram-positive bacteria and fungi. Please clarify.
- In the method section, subpart Microbiome Analysis, it appears there may be a copy-paste error in the primer sequences. The sequence listed for the 806R reverse primer is identical to the 341F forward primer. Please clarify and correct.
- The manuscript states that a combined 16S/18S microbiome analysis was performed to identify both bacteria and fungi. However, the methods only detail the use of universal prokaryotic primers (341F/806R) that target the 16S rRNA gene. There is no mention of the specific primers or protocol used to amplify the fungal 18S rRNA gene. Please clarify and correct.
- The visual presentation of Figure 2 could be enhanced because the numbers representing the total reads above each bar are quite small and difficult to read
- The bar charts in Figure 1 clearly show that a very large proportion of the sequencing reads were from the host. This is a common and important challenge in microbiome studies. It would strengthen your discussion to address the implications of this. For instance, you could discuss how high levels of host DNA can mask the presence of low-abundance pathogens by consuming sequencing depth, and perhaps mention potential future strategies to mitigate this, such as host DNA depletion kits
- It was very interesting to note that over a third of your patient cohort was positive for SARS-CoV-2. Please add a brief comment in the discussion about whether you observed any notable patterns or differences in the microbiomes of the COVID-19 positive patients would be a timely and valuable addition
- In your discussion, you effectively compare your findings to a study by Heitz et al. that used metagenomic next-generation sequencing (mNGS). Since your study used 16S/18S amplicon-based sequencing, it could be better to include a sentence or two briefly explaining the difference.
- The introduction effectively describes the challenges in diagnosing Nosocomial Pneumonia (NP). To strengthen it further, you could more explicitly state the clinical dilemma that arises when a patient has clear signs of pneumonia, but cultures are negative, often due to prior antibiotic use.
- You correctly introduce 16S/18S sequencing as a method that doesn't require viable bacteria. Consider adding a brief explanation of its other advantages, such as its ability to detect fastidious organisms
- Your choice to define a pathogen as an organism contributing more than 40% of sequence reads is a critical methodological decision. Please expand your discussion on the rationale for this specific cutoff.
- The finding that 47.8% of samples had no microorganisms identified by microbiome analysis needs more explanation. Please clarify if this means no non-vertebrate DNA was detected, or simply that no single organism met the 40% threshold.
- Your study successfully identified unusual pathogens like Pasteurella canis and Tropheryma whipplei. The discussion would be significantly strengthened by briefly mentioning the clinical impact of these findings.
- You rightly touch upon this issue when discussing the Tropheryma whipplei This is arguably the biggest challenge for highly sensitive molecular tests. Expanding this into a broader discussion point would be a valuable addition.
- The conclusion appropriately calls for further studies to refine diagnostic thresholds
Author Response
- To immediately frame the scale of this work for the reader, you might consider adding the total number of patients (N=23) to the methods section of the abstract.
We have moved the number of patients to the first line i.e. 26 from line 27 to make it immediately apparent to the reader (line 26-27).
- The manuscript lacks any information regarding the handling of samples after collection. Key details such as the maximum time allowed between collection and processing, and the storage temperature (e.g., 4°C, -80°C) during that interval should be added.
We have expanded the methods section regarding sample collection (line 139-142)
- The protocol specifies a kit-based DNA extraction method but fails to mention whether a mechanical lysis step was included. Standard enzymatic lysis is notoriously inefficient for breaking open organisms with tough cell walls, particularly Gram-positive bacteria and fungi. Please clarify.
We have updated, specified and corrected the protocols for DNA extraction (line 156-160)
- In the method section, subpart Microbiome Analysis, it appears there may be a copy-paste error in the primer sequences. The sequence listed for the 806R reverse primer is identical to the 341F forward primer. Please clarify and correct.
We have corrected and added the primer sequences (line 165-171)
- The manuscript states that a combined 16S/18S microbiome analysis was performed to identify both bacteria and fungi. However, the methods only detail the use of universal prokaryotic primers (341F/806R) that target the 16S rRNA gene. There is no mention of the specific primers or protocol used to amplify the fungal 18S rRNA gene. Please clarify and correct.
Please see the added primers in line 165-171.
- The visual presentation of Figure 2 could be enhanced because the numbers representing the total reads above each bar are quite small and difficult to read.
We have tried to optimize the font size for figure 2.
- The bar charts in Figure 1 clearly show that a very large proportion of the sequencing reads were from the host. This is a common and important challenge in microbiome studies. It would strengthen your discussion to address the implications of this. For instance, you could discuss how high levels of host DNA can mask the presence of low-abundance pathogens by consuming sequencing depth, and perhaps mention potential future strategies to mitigate this, such as host DNA depletion kits
We have discussed this issue in line 313-318.
- It was very interesting to note that over a third of your patient cohort was positive for SARS-CoV-2. Please add a brief comment in the discussion about whether you observed any notable patterns or differences in the microbiomes of the COVID-19 positive patients would be a timely and valuable addition
We have added a statement in the discussion detailing our findings in the discussion. (line 288-290)
- In your discussion, you effectively compare your findings to a study by Heitz et al. that used metagenomic next-generation sequencing (mNGS). Since your study used 16S/18S amplicon-based sequencing, it could be better to include a sentence or two briefly explaining the difference.
We have added a statement in the discussion 313-318
- The introduction effectively describes the challenges in diagnosing Nosocomial Pneumonia (NP). To strengthen it further, you could more explicitly state the clinical dilemma that arises when a patient has clear signs of pneumonia, but cultures are negative, often due to prior antibiotic use.
We have elaborated on this dilemma in the introduction (line78-80).
- You correctly introduce 16S/18S sequencing as a method that doesn't require viable bacteria. Consider adding a brief explanation of its other advantages, such as its ability to detect fastidious organisms
We have added a sentence elaborating this (Line87-90).
- Your choice to define a pathogen as an organism contributing more than 40% of sequence reads is a critical methodological decision. Please expand your discussion on the rationale for this specific cutoff.
As we described in the manuscript (lines 160-163), there is no reference methodology for NGS based diagnostics of NP. Therefore, we chose a proportion-based approach identifying clinically relevant pathogens. Specifically, pathogens contributing more than 40% of the total sequence reads were considered clinically significant and reported. A minor revision detailing was performed to clarify our message.
- The finding that 47.8% of samples had no microorganisms identified by microbiome analysis needs more explanation. Please clarify if this means no non-vertebrate DNA was detected, or simply that no single organism met the 40% threshold.
We added “In 22 samples, no microorganisms were identified. In 12 of these, no microbial DNA was detected, while in the remaining 10 samples, the microorganisms present did not meet the 40% abundance threshold for reporting.” (lines 232-234).
- Your study successfully identified unusual pathogens like Pasteurella canis and Tropheryma whipplei. The discussion would be significantly strengthened by briefly mentioning the clinical impact of these findings.
We have stated the interpretation of the clinical impact in lines (335-341) and (355-357).
- You rightly touch upon this issue when discussing the Tropheryma whipplei This is arguably the biggest challenge for highly sensitive molecular tests. Expanding this into a broader discussion point would be a valuable addition.
We expanded the discussion in lines 330-335 with two new references.
- The conclusion appropriately calls for further studies to refine diagnostic thresholds
Reviewer 3 Report
Comments and Suggestions for Authors
To: Diagnostics MDPI
Dear EIC,
Dear AE,
I hope you are doing well.
This is my review report for the manuscript ID: diagnostics-3928976.
This study evaluates the diagnostic accuracy of microbiome analysis and culture methods for diagnosing BAL and TS samples in intubated, critically ill patients with nosocomial pneumonia admitted to ICUs. These two sites are sterile body sites. The cultured pathogens were identified using MALDI-TOF MS. The microbiome analysis was performed by PCR and sequencing of 16S/18S rRNA genes. They concluded that molecular identification reached a high accuracy. The results of the study do not add anything novel to the field. I listed my comments below.
Comments
- Please check the manuscript for writing and grammatical errors. Line 101
- Why did the authors select this as an exclusion criteria? “being intubated for more than 24 hours before screening”
- There is a low number of cases (23 patients) in this study. This can affect the value of the results.
- Cultures may carry environmental pollution risks, and it is clear that molecular identification has higher accuracy. However, I wonder why the authors extracted DNA from cultured microbes. There is a risk of environmental contamination of cultures. The DNA should be extracted directly from clinical samples, as the aim of this study is to assess diagnostic accuracy.
- In the conclusion section, the authors dismiss their own results, as they mentioned in the results section of the abstract. They state that their findings are consistent with both culture and molecular methods and conclude that these methods should complement each other. I read the manuscript and noticed that the diagnostic accuracy of the microbiome analysis is higher than that of culture methods. How do the authors justify this?
- Low number and quality of the listed references.
Good Luck
Author Response
- Please check the manuscript for writing and grammatical errors. Line 101
We have revised this paragraph so that the sentences are more grammatically correct (lines 104-107). Furthermore we have looked the manuscript through and made minor adjustments if needed.
- Why did the authors select this as an exclusion criteria? “being intubated for more than 24 hours before screening”
The limit to the length of intubation was based on the risk of biofilm formation in the endotracheal tube and risk of including colonization in the analyses. This information is added in lines 120-124.
There is a low number of cases (23 patients) in this study. This can affect the value of the results.
We added the sentence “The descriptive nature of the study, along with the low number of cases included, represents a limitation to the strength of the conclusions that can be drawn.” (lines 366-367).
- Cultures may carry environmental pollution risks, and it is clear that molecular identification has higher accuracy. However, I wonder why the authors extracted DNA from cultured microbes. There is a risk of environmental contamination of cultures. The DNA should be extracted directly from clinical samples, as the aim of this study is to assess diagnostic accuracy.
We have elaborated on this in the methods section. The DNA extraction was not from the cultured isolates, but directly from the original sample material (line 139-142).
- In the conclusion section, the authors dismiss their own results, as they mentioned in the results section of the abstract. They state that their findings are consistent with both culture and molecular methods and conclude that these methods should complement each other. I read the manuscript and noticed that the diagnostic accuracy of the microbiome analysis is higher than that of culture methods. How do the authors justify this?
We concluded from the study that microbiome analysis does not have higher diagnostic accuracy than culture. It is a good supplement to the conventional culture. By combining both methods could enhance the diagnostic accuracy for NP. We added the sentence “Combining microbiome analysis with traditional culture techniques could enhance the diagnostic accuracy for NP.” in the abstract to clarify this (lines 44-45).
- Low number and quality of the listed references.
We have added 8 new references to expand and improve the reference section.
Reviewer 4 Report
Comments and Suggestions for Authors
Dear Authors,
I have reviewed your article entitled “Comparison of Cultures and 16S/18S Amplicon-Based Microbiome Analyses for Diagnosing Nosocomial Pneumonia in Patients Admitted to the Intensive Care Unit – An Exploratory Study”.
This is a prospective, descriptive cohort study aimed at comparing diagnostic methods in patients with nosocomial pneumonia (NP) in the intensive care unit. The diagnostic concordance between culture results from tracheal aspirate (TS) and bronchoalveolar lavage (BAL) samples and 16S/18S rRNA gene amplicon-based microbiome analyses was examined. The study highlights the limitations of classical culture methods and demonstrates the potential contribution of sequencing-based analyses. Therefore, your study is valuable.
However, some revisions should be made:
- Although the number of patients is small, further statistical analysis would enhance the study and should be conducted.
- The high concordance between TS and BAL was noted, but further statistical analysis is necessary to calculate concordance.
- A detailed evaluation of treatment changes is also recommended. The impact of NGS on treatment changes should be stated.
- Although intubation for more than 24 hours was among the exclusion criteria, the average intubation time for the patients was stated as 11 days. This is unclear.
- The SARS-CoV-2 positivity rate reported in the demographic data in the 'Results' section is unclear. Were these patients admitted to the intensive care unit due to COVID-19, or were they detected positive because routine SARS-CoV-2 PCR testing was performed on every patient admitted to the intensive care unit? It is recommended that these patients be excluded from the study if they were admitted to the intensive care unit due to COVID-19.
- Some bacterial names are not in italics. This should be corrected.
- For reference 11, it is recommended to use this link: https://www.cdc.gov/nhsn/pdfs/pscmanual/6pscvapcurrent.pdf.
Best regards,
Author Response
- Although the number of patients is small, further statistical analysis would enhance the study and should be conducted.
The descriptive nature of the study, along with the low number of cases included, represents a limitation to the strength of the conclusions that can be drawn.
We added this sentence at the end of the discussion to address this limitation (lines 361-362).
- The high concordance between TS and BAL was noted, but further statistical analysis is necessary to calculate concordance.
The descriptive nature of the study, along with the low number of cases included, represents a limitation to the strength of the conclusions that can be drawn.
We added this sentence at the end of the discussion to address this limitation (lines 361-362).
- A detailed evaluation of treatment changes is also recommended. The impact of NGS on treatment changes should be stated.
This is a prospective study and the cultures were performed “real time” after sampling. However, the microbiome based on NGS was performed more than 3 months after the samples were collected. Therefore, the results of the NGS analysis did not have influence on treatment.
Although intubation for more than 24 hours was among the exclusion criteria, the average intubation time for the patients was stated as 11 days. This is unclear.
Samples were collected within 24 hours after intubation to avoid the risk of biofilm formation in the endotracheal tube and risk of including colonization in the analyses. This limit for sample collection does not reflect the duration of ventilator treatment.
- The SARS-CoV-2 positivity rate reported in the demographic data in the 'Results' section is unclear. Were these patients admitted to the intensive care unit due to COVID-19, or were they detected positive because routine SARS-CoV-2 PCR testing was performed on every patient admitted to the intensive care unit? It is recommended that these patients be excluded from the study if they were admitted to the intensive care unit due to COVID-19.
We did not find any discernible differences between the group that tested positive for SARS-COV 2 and the group that tested negative. (lines 288-290)
- Some bacterial names are not in italics. This should be corrected.
The document has been thoroughly looked through and corrections have been made where necessary.
- For reference 11, it is recommended to use this link: https://www.cdc.gov/nhsn/pdfs/pscmanual/6pscvapcurrent.pdf.
This reference has been updated.
Round 2
Reviewer 3 Report
Comments and Suggestions for Authors
The reviosion version was checked. It is approved for further editorial desicion.
Reviewer 4 Report
Comments and Suggestions for Authors
Dear Authors,
Thank you for making the necessary revisions and clarifying some issues.
Best regards,